# Three forms of temporal disorientation: A thematic analysis of subjective reports about Covid-19 restriction periods

**Bastien Perroy**[1☉]**, Pablo Fernandez Velasco**[2☉*]**, Umer Gurchani**[1]**, Roberto Casati**[1]

**1** Institut Jean-Nicod (ENS, EHESS, CNRS), Paris, France, **2** Centre for the Sciences of Place and Memory, University of Stirling, Stirling, United Kingdom

☉ These authors contributed equally to this work.
* p.fernandezvelasco@gmail.com

## Abstract

During the Covid-19 restrictions, people reported various surprising disruptions in their experience of time, such as time simultaneously passing slower and faster, or feeling unreal. This hints at instances of dissociative time experiences in the general public in times of crisis. We investigate the temporal experience of the pandemic through a corpus-based thematic analysis and a multiple correspondence analysis of 149 subjective reports gathered in March 2021, during a period of long lasting and ongoing restrictions in France and the UK. We argue that three forms of temporal disorientation constitute a fitting umbrella over a heterogeneous phenomenology. The loss of temporal landmarks made it harder to orient oneself and induced episodic forms of temporal disorientation. Distinctively, sustained temporal disbelief, an existential form of temporal disorientation, could occur when people's past perspective was severely distorted. Finally, a future-oriented form of temporal disorientation whose hallmarks were feelings of anxiety and hopelessness could occur alongside inabilities to project oneself into the future. Our findings suggest that future landmarks should be provided to those most exposed to dissociative temporal experiences during crises.

## Introduction

The Covid-19 pandemic has resulted in an unprecedented and uncanny temporal experience for many individuals. The New York Times termed 2020 *The Year of Blur* [1]: time was "without boundaries", a "momentous year" during which it was hard for people "to anchor themselves in reality". These exceptional times superseded the previous temporality in confusing ways. The sudden need for confinement measures was a disturbing surprise for societies at large. Additionally, a metaphorical mental curtain impaired our ability to feel as confident as before about our near future. Accordingly, the Covid-19 pandemic has been described as a source of both

**Data availability statement:** The data (both unprocessed and processed, including intercoder reliability and MCA metrics) can be explored in this repository: https://doi.org/10.17605/OSF.IO/H67NF

**Funding:** This work was supported by the Agence Nationale de la Recherche under Grant Agreement Numbers ANR-17-EURE-0017 (FrontCog) and RA-COVID-19 V11 (DISCovid). BP benefited from a grant from RATP Group for a research project on temporal disorientation. PFV benefited from a postdoctoral fellowship from the British Academy (PFSS23\230053). The funders had no role in study design, data collection and analysis, decision to publish, or preparation of the manuscript.

**Competing interests:** No potential conflict of interest was reported by the author(s).

individual and social trauma [2,3]. One study found that 36% of participants across multiple countries experienced PTSD symptoms related to the pandemic [4]. New local epidemic outbreaks were accompanied with continuously evolving mitigation strategies. For example, in countries like France, lockdowns transitioned into various forms of curfews throughout 2020 and 2021. These changes contributed to making the experienced time distortions persist throughout the period [5,6]. Years after the onset of the pandemic, this crisis remains a fruitful object of study. It offers insights into how acute crises, including political and economic upheavals, can disrupt lived temporalities [7].

A blossoming literature found that the temporal phenomenology during the pandemic varied across individuals or social contexts. Studies have reported opposite findings regarding the perceived speed of time passage, with some indicating it felt faster [8,9] and others slower [10,11]. Researchers have noted puzzling perceptions about temporal experiences. For instance, some individuals reported time feeling elongated while also passing faster, with statements such as "*it has already been x weeks since the beginning of the quarantine*" coupled with "*it feels as if it started a year ago*" [12]. The most noteworthy international enterprise to date has been *The Blursday Database* [13], which established the prime role of perceived isolation in the experienced distortions. Several studies have highlighted the significant role of moods, such as boredom [10,14] and depression [6], in shaping temporal phenomenology during the crisis. Likewise, another study [15] provided quantitative evidence of significant positive associations between COVID-19-related trauma and symptoms of anxiety, depression, and insomnia. These findings further illuminate the interconnected nature of pandemic-induced psychological distress and temporal experiences. However, more work is still needed to provide a comprehensive taxonomy of the heterogeneous temporal distortions experienced after such a traumatic societal event.

Our team wanted to study the emotional impact of the crisis by focusing on its temporal and social disorienting aspects. We used spatial disorientation as our baseline, as it is scientifically better understood than temporal or social disorientation. Spatial disorientation is a metacognitive feeling which contributes to the subject's monitoring of her navigation [16]. Disorientation increases epistemic vigilance when cognitive maps encoding for the surrounding spatial environment break down. Many orientation researchers suggest that some cognitive features of spatial orientation may be structurally shared across domains [17–19]. Conversely, the partial neural overlapping of orientational activity across domains also entails important dissimilarities. Some researchers suggest that self-involving features of orientation, such as egocentric and allocentric mapping and projection onto abstract reference planes might be the essential shared mechanism across orientational domains [20]. Accordingly, disorientation experiences can be framed as dissociative insofar as they hint at a breakdown of the integration of the self across lived dimensions. We hypothesized that disruptions of temporal and social landmarks during Covid-19 made people feel socially and temporally disoriented, proper [21–25]. In psychopathology, disorienting shocks are described as breaking the "affective-conative momentum" of lived non-reflexive time [26]. Before the shock, time was experienced as an implicit temporal continuum

where one could act affectively without mental effort. After the shock, this temporality is replaced by a dissociated and reflexive experience of time. This new experience is fractured along past, present, and future perspectives, each associated with sets of orientation-specific emotion. To our knowledge, no study has examined the interplay between these dissociative time experiences during the Covid-19 crisis and their associated emotions.

Our aim is twofold. Firstly, we provide a comprehensive, corpus-based taxonomy of affective temporal experiences associated with the Covid-19 crisis. To achieve this, we created and shared an open-ended questionnaire. Secondly, we clarify the central experiential aspects, arguing that they comprise three forms of temporal disorientation, each aligned with a different time perspective. In the following sections, we describe a thorough thematic analysis and Multiple Correspondence Analysis (henceforth, MCA) of 149 subjective reports. These reports were gathered from French and English respondents in March 2021, during a prolonged period of heavy Covid-19 restrictions.

## Materials and methods

### Study design: thematic analysis based on an open-ended questionnaire

Previous studies have successfully used thematic analysis to generate insights into the phenomenological experience of disoriented individuals, such as those with dementia [27]. Various forms of thematic analysis exist in the literature [28]. In this study, we adopt a quantitative or reliability-based approach to thematic analysis of subjective reports. This approach allows us to focus on the associations between the various phenomena that people express regarding their experience of time during confinement measures. Our study design combines qualitative thematic analysis with quantitative Multiple Correspondence Analysis (MCA). This mixed-methods approach allows us to not only identify themes in subjective reports but also to quantitatively analyze the relationships between these themes, providing a more comprehensive understanding of temporal distortions during the Covid-19 crisis.

Based on established best practices and examples from the medical literature [29], we modeled an open-ended questionnaire by framing our question indirectly while providing examples. Disorientation can sometimes be framed as a pleasant affective experience despite its overall negative valence [30]. Additionally, anecdotal evidence of potential upsides to lockdown experiences was occasionally expressed in the public sphere. Given these factors, we decided to include both positive and negative examples in our questionnaire:

Some people have been feeling time distortions as a result of the Covid-19 health crisis. To name but a few examples:

- You may have felt a change in how quickly or slowly time passes,

- you may have felt unable to project yourself into the future or relieved at not having to make any plans,

- you may have been unsure of what day of the week or the month it was,

- you may feel that past events are further or nearer into the past than they actually are,

- or you may have found yourself following more natural times and schedules than those imposed from work or education.

Please use the space below to narrate any ways in which you might have felt time distortions as a result of the current health crisis. Feel free to write down anything you feel is relevant (e.g., an episode that struck you particularly).

### Sampling procedure: convenience sampling through university mailing lists

We adopted a convenience sampling strategy. In March 2021, we distributed the questionnaire through several university mailing lists in France and in the UK to which we had access. These mailing lists were general, from undergraduate to university employees and faculty. This sampling procedure was motivated by the fact that university students were an at-risk population from a mental health perspective during the crisis [31] and that we wanted to gather timely data on the ongoing restrictions. At the time respondents filled the questionnaire, France was undergoing a 3 month-long

curfew (which at its worst, forbade people to be outside after 6PM) and that was preceded by the 7 weeks of the second French national lockdown of October 2020. The pattern of restrictions in the UK was similar, with a second national lockdown in November 2020, a third national lockdown in January 2021 and regional tiered restrictions in between. In both contexts, respondents had experienced heavy restrictions over a 4–5 months period. The Pôle Éthique of the Institut des Sciences Biologiques (INSB) of the Centre national de la recherche scientifique (CNRS) waived all ethical approval for fully anonymous questionnaires. The study has been conducted according to the principles expressed in the Declaration of Helsinki. Our target sample size was based on a balance between achieving sufficient statistical power for detecting strong associations in our coding procedure and maintaining a manageable workload for our labour-intensive qualitative analysis. We aimed for a minimum of 100 responses, as this number would provide adequate data for our analytical approach while still allowing for thorough, in-depth coding of each response. We set an upper limit of a few hundred responses to ensure the qualitative analysis remained feasible within our research constraints. In total, 149 respondents consented to participate in our study. Table 1 details our sample characteristics, which ended up being relatively diversified. Average and median age across the French and English distributions were similar but much higher than expected for a student population, suggesting a higher response rate among education professionals.

### Coding procedure: establishing a taxonomic tree and delimiting units of meaning

Our two researchers specializing in disorientation first read the reports multiple times. They then independently tagged and commented on the recurrent themes they observed emerging. After meeting to discuss the relevant themes, the researchers agreed on a taxonomic tree consisting of 40 tags.

The taxonomic tree was composed of four overarching set of tags: tags related to time distortions (e.g., faster passage of time), tags related to temporal agentive feelings (e.g., time feeling unreal), tags related to disorientation (e.g., being lost in time) and temporal horizons tags (e.g., future). Temporal horizon tags would allow for further fine-grained analyses. We expected to observe different distortions at various scales (e.g., hour of the day for a day-scale, day of the week for a week-scale), across different time-oriented perspectives (e.g., past, present, future), and during specific periods (e.g., lockdown, curfew, "before Covid").

Each tag was framed as a dichotomous yes/no variable, as many reports described a given distortion (e.g., passage of time) in multiple ways (e.g., feeling time passing both faster and slower simultaneously). We divided the 149 reports into 355 units of meaning. This process involved retrieving the tagged sections from the coder specializing in temporal disorientation, removing the tags, and coherently dividing the remaining report content into meaningful units. We used Excel as our labelling software to accommodate the complex logic of our 3-step tagging process.

**Table 1. Sample characteristics.**

| Characteristics | Value |
| --- | --- |
| Number of respondents | 149 |
| Respondents of the French questionnaire | 95 (64%) |
| Respondents of the English questionnaire | 54 (36%) |
| Average age | 37.9 |
| Median age | 33.5 |
| Female respondents | 102 (68%) |
| Male respondents | 43 (29%) |
| Respondents that identified neither 'male' or 'female' | 4 (3%) |
| Respondents located neither in France nor the UK | 28 (19%) |

## Reliability assessment: a 3-step tagging process

The tagging process consisted of three rounds. In the initial blind tagging round, two coders independently assessed each of the 355 units of meaning against all 40 tags, resulting in 14200 taggable instances. After the first round, intercoder agreement was at 96.4% including null agreements and 50.9% when excluding them, out of 1,033 attributed tags. These ratios are consistent with literature standards. The difference in intercoder reliability with and without null agreements can be attributed to the cognitive load placed on coders, who must continuously monitor numerous tags over several hours of tagging [32].

To assess intercoder reliability accounting for change agreement, we used Cohen's κ and Krippendorff's α [33]. Cohen's κ compares observed agreement to the hypothetical probability of chance agreement, while Krippendorff's α compares observed disagreement to expected disagreement. The mean average of Cohen's κ across the 40 tags was 0.605 (SD = 0.17), and Krippendorff's α was 0.607 (SD = 0.17). At this stage, these rates were substandard and showed substantial standard deviation. This indicated that at least some tags had unsatisfactory intercoder agreement, prompting us to engage into a round of negotiated agreement [34].

This second round consisted in blind revision, in which each coder could see the differences, add new tags, or retract previously attributed ones. Importantly, the two coders conducted their new tagging independently. This meant that the second round could have potentially worsened the agreement rate. For example, if one coder added a tag based on the other coder's initial judgement, while the other coder simultaneously retracted that tag. This round made the agreement rate rise to 99.3% including null agreements and 90.3% excluding null agreements. The average Cohen's κ across tags rose to 0.932 (SD = 0.06), and the average Krippendorff's α also rose to 0.932 (SD = 0.06). Both the arithmetic means and standard deviations of these metrics improved substantially. Importantly, no tag had a Cohen's κ or a Krippendorff's α lower than 0.77, demonstrating consistent reliability across all available tags. Most revisions involved convergence (n = 394) rather than retraction (n = 77). This sharp increase in agreements during the second round suggested that most of the previous disagreements were likely due to the cognitive load associated with the tagging process. As these rates were substantially above recommended standards [35] we moved to the final round, in which both coders met and discussed the few remaining disagreements to generate a final database (table 2).

## Statistical tests: using McNemar's test to assess non marginal homogeneity

We excluded units of meaning tagged "unrelated to Covid" (n = 10), which referred to disruptions predating Covid-19 or associated with external events (e.g., childbirth). We also included dichotomous variables for the 10 themes that constitute our taxonomic tree, which we review in the discussion section. Each unit of meaning was assigned these new tags if it had been previously tagged with at least one corresponding tag in these themes or meta-themes. Adding these variables allowed us to examine frequency correlations across multiple levels, such as between a theme and a tag in an opposing theme. Both the mean and median number of tags per report (for reports with at least one tag) was 6, suggesting potentially interesting correlations among frequently co-occurring tags.

**Table 2. Evolution of intercoder agreement across rounds and branches of the taxonomic tree.**

| Round | 1 (blind tagging) | | | | 2 (blind revision) | | | | 3 |
|---|---|---|---|---|---|---|---|---|---|
| Metric | I.C agree. | I.C agree. excl. null. | average Cohen's κ | average Krippendorff's α | I.C agree. | I.C agree. excl. null. | Average Cohen's κ | average Krippendorff's α | I.C agree. |
| Total | 96.4% | 50.9% | 0.607 | 0.605 | 99.3% | 90.3% | 0.932 | 0.932 | 100% |
| Time distortions | 97.1% | 62.1% | 0.654 | 0.654 | 99.1% | 88.4% | 0.912 | 0.912 | 100% |
| Temporal agentive feelings | 98.4% | 50.0% | 0.599 | 0.599 | 99.5% | 83.3% | 0.923 | 0.923 | 100% |
| Disorientation related states | 93.3% | 33.0% | 0.455 | 0.451 | 98.7% | 87.3% | 0.930 | 0.930 | 100% |
| Temporal horizons | 94.2% | 42.8% | 0.579 | 0.575 | 99.5% | 95.6% | 0.972 | 0.972 | 100% |

We analyzed 1114 2×2 contingency tables excluding pairs of the same variable, redundant pairs, as well as pairs of overlapping phenomena (such as a theme and a tag within that theme). Our goal was to compute the phi correlation coefficient ($r_\varphi$). We started by assessing significant relationships with the McNemar's test of marginal homogeneity. We did not use alternatives such as the $\chi^2$ test of independence or Fisher's exact test. The former was unsuitable due to our need for an exact test, while the latter was inappropriate for our paired data.

Several versions of the McNemar's test exist. We chose the exact mid-p value version for two reasons. Firstly, the standard non-exact version of the test requires that the sum of discrepant values for each pair shouldn't be less than 25. 15.3% of our contingency tables didn't fit this condition. Furthermore, a recent meta-analysis found the mid-p value to be the most reliable version across contexts, recommending its default use [36]. To prevent type I error inflation, we applied a correction technique for all comparisons. Given the high number of tests in our matrix, we focused on controlling the proportion of false positives among our discoveries, rather than the overall probability of any false positive error. We applied the Benjamini-Hochberg correction to control the false discovery rate while maintaining reasonable statistical power. Out of our 1114 correlations, 645 had a p-value below 0.01, 90 had a p-value between 0.01 and 0.05, and 379 had a p-value above 0.05. Setting our α threshold at 0.05, we rejected the null hypothesis (of marginal homogeneity in the contingency table) for 735 pairs, representing 66% of the total matrix. This allows us to investigate significant correlations with noteworthy effect sizes (or at least small effects, with $r_\varphi > 0.2$) in the results section.

### Interpretative instrument: performing MCA to synthesise the disorientation structure

Multiple Correspondence Analysis (MCA) is a dimensionality reduction technique specifically designed for categorical variables [37]. In case of annotated textual data, such as ours, it can provide a coherent visual summary of our results as MCA compresses all the variables of a dataset into dimensions that maximize explained inertia, revealing underlying structures by preserving as much variance as possible in the process. Each of these new dimensions are a blend of variables (i.e., tags) to which each variable contributes more or less and onto which both individuals (i.e., units of meaning) and variables can be projected. Epistemologically, MCA permits to study the data *all things included* rather than *all else being equal*: disorientation being a highly contextual phenomenon, this holistic approach to the data seemed relevant. We chose to integrate the 10 themes and meta-themes tags as supplementary variables, which means that they aren't integrated in the computation that results in these new dimensions but nonetheless can be projected onto them. This choice allowed us to see whether the way we theoretically enrich our taxonomic tree through thematic analysis is congruent with how the dataset behaves when we reduce its dimensions.

## Results

### Taxonomic tree and significant relationships between tags

The full taxonomic tree with the number of attributed tags is shown in Fig 1. We used McNemar's mid-p value test for significance testing and r φ for effect sizes, as detailed in the methods section. All statistical tests performed are available in the data archive linked to our study.

As can be seen in table 3, three interesting sets of significant frequency correlations between tags stand out within our data: perspective-dependent correlations, temporal scale or landmark dependent correlations, and correlations in between themes or tags unrelated to temporal horizon.

Future-oriented units of meaning display the strongest correlation within our dataset with difficulties to project oneself in the future (r φ=0.89, p=0.012). Interestingly, feeling stuck in the present seems to involve both present (r φ=0.30, p<0.001) and future orientation, or noticeable lack thereof (r φ=0.22, p<0.001). The past is mainly characterized by the "subjective temporal distances" theme, as seen in table 4.

Different temporal scales or landmarks are associated with different temporal disruptions, as shown in table 5. The strongest associations take place on the one hand on the week-scale and temporal orientation tags, on the other with the

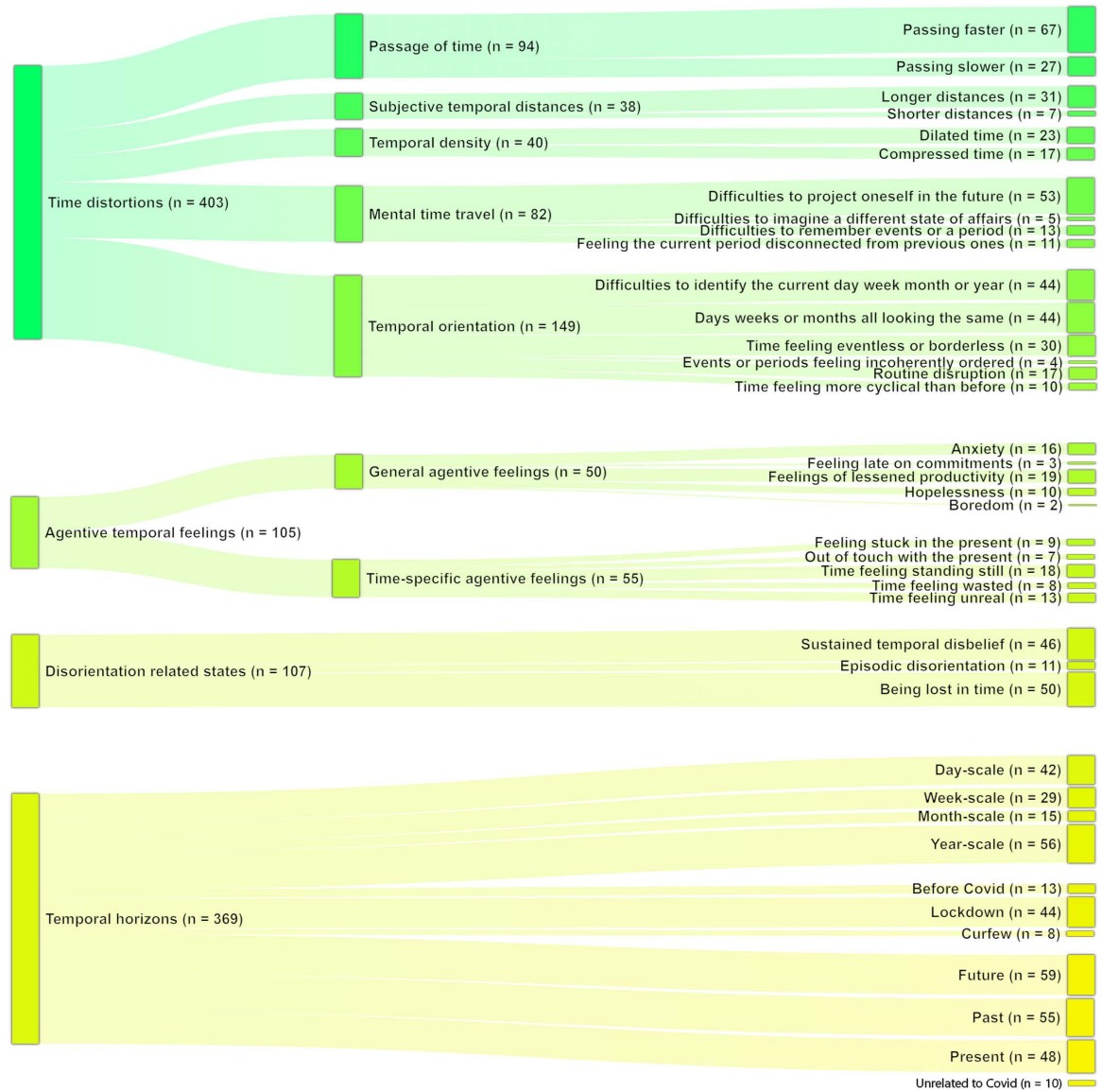

**Fig 1. Sankey diagram of the taxonomic tree resulting from thematic analysis.**

"before covid" temporal landmark which shows a noteworthy correlation with temporal distances (r φ = 0.44, p < 0.001). Disorientation related states seem to slightly intervene more in between the week-scale (r φ = 0.28, p < 0.001) and year-scale (r φ = 0.23, p < 0.001). The day-scale is associated with routine disruption (r φ = 0.21, p < 0.001), time feeling cyclical (r φ = 0.20, p < 0.001) as well as compressed time (r φ = 0.20, p < 0.001). The month-scale (i.e., with a week of the month or day of the month framing) didn't appear as a structuring temporal scale in our reports.

Alterations of temporal orientation show a relatively strong correlation with being lost in time (r φ = 0.56, p < 0.001). In contrast with episodic disorientation being associated with temporal orientation (r φ = 0.29, p < 0.001), sustained disbelief shows a slight correlation with time feeling unreal (r φ = 0.33, p < 0.001). Hopelessness is associated with difficulties to project oneself in the future (r φ = 0.23, p < 0.001).

**Table 3. Perspective-dependent significant correlations.**

| Tag 1 | Tag 2 | r φ | p value |
|---|---|---|---|
| Future | Difficulties to project oneself in the future | 0.89 | 0.012 |
| Future | Alterations of mental time travel | 0.73 | 0.001 |
| Future | Hopelessness | 0.25 | 0.000 |
| Future | Feeling stuck in the present | 0.22 | 0.000 |
| Present | Feeling stuck in the present | 0.30 | 0.000 |
| Past | Subjective temporal distances | 0.43 | 0.003 |
| Past | Difficulties to remember events or a period | 0.29 | 0.000 |
| Past | Time feeling unreal | 0.25 | 0.000 |
| Past | Shorter distances | 0.22 | 0.000 |

**Table 4. Temporal scale or landmark dependent significant correlations.**

| Tag 1 | Tag 2 | r φ | p value |
|---|---|---|---|
| Day-scale | Routine disruption | 0.21 | 0.000 |
| Day-scale | Compressed time | 0.20 | 0.000 |
| Day-scale | Time feeling more cyclical than before | 0.20 | 0.000 |
| Week-scale | Difficulties to identify the current day, week, month or year | 0.46 | 0.029 |
| Week-scale | Being lost in time | 0.45 | 0.002 |
| Week-scale | Temporal orientation | 0.34 | 0.000 |
| Week-scale | Disorientation related states | 0.28 | 0.000 |
| Year-scale | Disorientation related states | 0.23 | 0.000 |
| Before Covid | Subjective temporal distances | 0.44 | 0.000 |
| Before Covid | Longer distances | 0.42 | 0.000 |

**Table 5. Significant correlations across themes and temporal disruptions.**

| Tag 1 | Tag 2 | r φ | p value |
|---|---|---|---|
| Alterations of temporal orientation | Being lost in time | 0.56 | 0.000 |
| Difficulties to identify the current day, week, month or year | Disorientation related states | 0.53 | 0.000 |
| Days, weeks or months all looking the same | Time feeling more cyclical than before | 0.35 | 0.000 |
| Time feeling unreal | Sustained temporal disbelief | 0.33 | 0.000 |
| Difficulties to identify the current day, week, month or year | Episodic disorientation | 0.29 | 0.000 |
| Difficulties to remember events or a period | Events or periods feeling incoherently ordered | 0.26 | 0.013 |
| Time distortions | Disorientation related states | 0.26 | 0.000 |
| Difficulties to project oneself in the future | Hopelessness | 0.23 | 0.000 |
| Being lost in time | Episodic disorientation | 0.21 | 0.000 |
| Time distortions | Being lost in time | 0.21 | 0.000 |
| Difficulties to remember events or a period | Time feeling eventless or borderless | 0.21 | 0.003 |
| Alterations of mental time travel | Hopelessness | 0.20 | 0.000 |

## Multiple correspondence analysis

The first and second dimensions of the MCA explain 6.91% and 6.68% of inertia respectively, or 13,59% combined. We obtain a relatively low percentage because we coded data on a wide taxonomic tree. Most of the information in our data pertains to the absence of tags (n = 12,604) rather than their presence (n = 968). Additionally, there are few strong

correlations; out of 760 significant correlations among 1,112 total, only 36 have $r_\varphi > 0.2$. This low explained inertia suggests that the captured phenomenology is heterogeneous, and MCA can only partially synthesize the inertia. The average of the sum of squared cosine over the first 2 dimensions and across the 39 active variables is 0.14. This score is quite low, mainly impacted by low squared cosine of variables found in the following themes and which are not going to be interpreted: "passage of time", "temporal density", "time-specific agentive feelings", and the "time distortions" meta-theme. These themes are the only ones showing substandard v-test values (between −1.96 and 1.96, equivalent to a p-value above 0.05). 51.3% of the v-test of active variables on these two first dimensions are above 1.96 (or below −1.96).

Overall, while the projection of variables and individuals onto the two first dimensions are interpretable, it is important to emphasise the partial nature of this interpretation and the heterogeneity of the explored phenomena. Variables (i.e., tags) are projected and filtered for low cos2 and for substandard v-test values. These filtered variables are then interpreted in the discussion section (Fig 2).

## Discussion

The Covid-19 pandemic suddenly constrained lifestyles, and throughout its tidal surges, people's experience of time became chaotic. People not only had to enter confinement, but also to adapt to new social (e.g., distancing), temporal (e.g., curfew) and spatial (e.g., travel restrictions) rules. This led to widespread societal confusion and uncertainty about ongoing events. We found that the temporal phenomenology of the Covid-19 crisis was best described as heterogeneous. It comprised three overarching meta-themes and three main forms of disorientation, corresponding to a wide variety of phenomena in our taxonomic tree. In this section, we illustrate our taxonomic tree while simultaneously providing theoretical support to it. Importantly, we introduce two noteworthy forms of disorientation: *present episodic confusion* and *sustained past-oriented disbelief*. Of these, only the latter is a dissociative experience. Secondly, our interpretation of the MCA results reveals an additional, future-oriented, form of disorientation. This form strongly involves temporal agentive feelings for which we did not have a specific tag. We conclude that experiences of temporal disorientation during the Covid-19 crisis could be of three distinctive forms depending on one's temporal horizon of experience, each being characterized by different temporal disruptions. Most noticeably, a dissociative form of temporal disorientation took the form of disbelief towards a year-scale past perspective and time feeling unreal.

### Unfolding the taxonomic tree

We identified 5 main time distortions, 2 groups of temporal agentive feelings, 2 forms of disorientation, and an underlying "being lost" state. We will discuss each of these in turn. Some of the reports that we quote have been translated from the original French.

### Time distortions

We report alterations related to passage of time, subjective temporal distances, temporal density, mental time travel, and temporal orientation.

*Passage of time.* Passage of time judgments are commonly distinguished from duration judgments [38]. In experimental contexts, disruptions of one have been observed without concurrent disruption of the other [39]. In our reports, no respondent mentioned disruptions of their abilities of making duration judgments. This does not necessarily imply that these abilities were not unaffected during the Covid-19 era (although a study found no such impairments [40]). Instead, it suggests that the more salient, disrupted aspect of one's engagement with time was the *feeling* of time's passage. One of the most salient distinctions between passage of time and duration judgments lies in their nature. The former is a feeling without any intentional object—that is, no specific entity it explicitly measures. The latter, conversely, is a direct

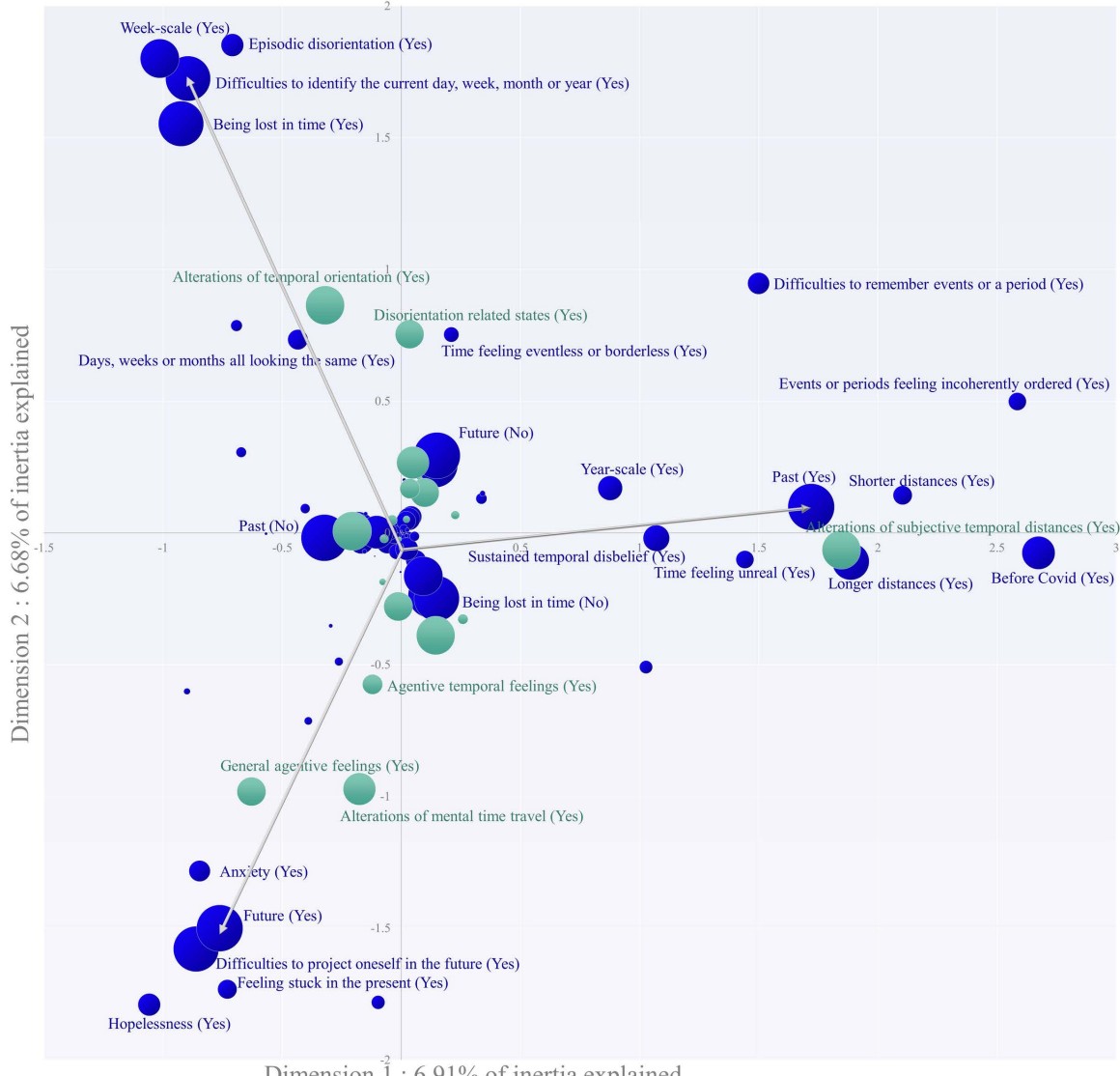

**Fig 2. Bubble chart of temporal disruptions one year into the Covid-19 crisis (MCA).** Primary variables are in blue, supplementary variables are in green. The surface area of bubbles is proportional to the squared cosine of the subtending tags which indicates representational quality. No labels are displayed for tags whose absolute average v-test value over the two dimensions was between −1.96 and 1.96, or for a sum of squared cosine inferior to 0.05 over the two dimensions. The three tags with the highest contribution value on either of the two dimensions are associated with a grey vector. Some poorly informational labels in the centre of the graph have been removed to ease visual cluttering.

measurement of a given quantity of elapsed time, whether prospective or retrospective. 11% of our tagged passage of time reports described both slower and faster feelings of time passage (e.g., "*time seems to pass slowly and quickly at once*", "*I have felt like time has passed both faster and slower*"]. This finding aligns with studies on time distortions during or about lockdowns, which reported distortions in both directions (e.g., [11,41]).

**Subjective temporal distances.** Subjective temporal distances refer to an intuitive feeling of the temporal distance between oneself and an event or a temporal landmark [13,42–44]. As intuitive feelings abouts temporal distances, they differ from passage of time judgments, which are concerned with the perceived flow of time passing. In our reports,

these distances were experienced in three ways: **longer** than their actual duration (e.g., "*it felt like 2020 took longer than most years to come to an end*"), **shorter** (e.g., "*the past year feels very short*"), or paradoxically both shorter and longer (e.g., "*The pandemic seems to be going on for both ten years and two weeks. It's difficult to conceptualize and uncanny.*"). Like passage of time, 9% of our tagged temporal distances reports were about both longer and shorter distances.

*Temporal density.* Both coders agreed on the phenomenological distinction between temporal density and subjective temporal distances when analyzing the reports and developing the taxonomic tree. However, existing literature does not appear to adequately illuminate this distinction. When people express **time compression** (e.g., "*bedtime seems to arrive much more quickly sometimes and you are hard put to understand what you have done to fill the last 15 hours*") or **time dilation** (e.g., " *Now, after a year in the pandemic, I feel like the pandemic time (that year) has an outsized position in my mental timeline*"), they appear to convey more than just longer or shorter temporal distances. These expressions seem to describe something more profound: they seem to state something about the fabric of time itself (rather than one's distorted subjective location within a normal fabric of time), e.g., "*I certainly felt both a time dilation and a time contraction: time is a spring.*". We do not assert that temporal density judgments have a different cognitive basis than subjective temporal distances. Rather, we propose that these expressions, although potentially metaphorical to some extent, are distinguishable from subjective temporal distancese.

*Alterations of mental time travel.* In our corpus, we identified four tags that encompassed most reports about mental time travel. This cognitive ability allows one to project oneself episodically in time (past or future) and to entertain counterfactual thoughts [45,46]. **Difficulties projecting oneself in the future** was the most prominent phenomenon (e.g., "*I can't project myself into the future because I don't even know what I'm going to do for the next thirty hours.*"). Closely associated were **difficulties to imagine a different state of affairs** ("*It seems that normal life will never come again*"; "*I feel bogged down, as if all of this (me unemployed, the closed facilities, the masks, the deaths, etc.) will never end. I know that's not true but it's a very strong feeling*"), **difficulties to remember events or a period** (e.g., "*I find I often can't remember what I did today, or yesterday*"), and **feeling the current period disconnected from previous ones** (e.g., "*everything that happened before the pandemic feels like it happened in some distant era, in the 'Before Times'*"). The central feature unifying these reports is an impairment of the decoupling process—the cognitive mechanism that allows the self to disengage from the present 'here and now'

*Alterations of temporal orientation.* We found six primary disruptions related to implicit temporal monitoring: **difficulties to identify the current day, week, month or year** (e.g., "*Sometimes I don't know where I stand during the week. I realize it for example when I want to go shopping when it is Sunday.*"), **days, weeks or months all looking the same** (e.g., "*Because of working from home, an impression that weekdays are no longer differentiated and that weekend days are just another day*"), **time feeling eventless or borderless** (e.g., "*All the months seem to be swallowed up into an undifferentiated whole because of the absence of birthday parties, family gatherings, holiday trips, etc*"), **events or periods feeling incoherently ordered** (e.g., "*The summer seems to me much more distant, ancient than the first confinement.*"), **routine disruption** (e.g., "*Loss of landmarks also in terms of circadian rhythm: getting up later, going to bed later, feeling of total disruption from that point of view...*") and **time feeling more cyclical than before** (e.g., "*Weeks or days seem similar and empty, feeling of eternal repetition.*").

Unlike the phenomena we previously categorized as mental time travel features, temporal orientation does not require decoupling. The self remains anchored in the present while attempting to determine its temporal location and context. These difficulties in basic temporal orientation were widely documented all throughout the pandemic. During the first lockdown, people were twice as likely to be confused about the day of the week compared to pre-lockdown periods [47]. In later stages of the crisis, individuals also experienced difficulties with prospective memory abilities. For example, they struggled to recall intentions such as "I'll call the doctor at 3PM" at the correct time, possibly due to working memory overload caused by increased anxiety and stress levels [48].

## Temporal agentive feelings

We distinguish time distortions from temporal agentive feelings because the former do not necessarily imply the distortion of one's agency, whereas the latter do.

*General agentive feelings.* The feeling grouped under "general agentive feelings" have been widely documented during the Covid-19 period in research outside the field of time perception. **Anxiety** was a recurring overarching emotion of the pandemic [49], as evidenced in many reports (e.g., "*I'm seriously stressed by the fact that I cannot know when the pandemic will end. Much of our pending activities depend on this fact, but the much-desired moment moves ahead and ahead*"). **Hopelessness** was documented for specific traumatically exposed populations (e.g., frontline healthcare workers facing death anxiety, [50]) and appears as one of the most striking disorientation phenomena. (e.g., "*loss of time and space - desire to disappear during confinement - fear of going out to reconnect. Is it always worth it*?"). **Boredom** substantially increased due to pandemic restrictions [51], which has been found to be a major predictor of a slowing of the passage of time [5,14]. However, we found only few explicit anecdotal evidence in our corpus (e.g., "*From the second lockdown onwards, the days seemed endless, boredom overwhelmed me, I could no longer work, no longer function.*"). **Feelings of lessened productivity** (e.g., "*I feel unable to project myself and I easily fall into a spiral of emptiness where I cannot do anything productive for several days. I feel very sluggish and unmotivated.*") and **feelings of being late on commitments** (e.g., "*During the first lockdown, I felt extremely rushed, like I didn't have enough time to reconcile everything.*") are two work-related salient phenomena in our corpus. These appear congruent with studies about feelings of productivity while remote working during lockdown [52].

All the aforementioned phenomena are general in the sense that none is time-specific. While they involve a temporal frame of reference (e.g., productivity relates to work over time, commitments are indexed to temporal "deadlines", hopelessness partly involves the veiling of one's future, and boredom partly relates to the sluggishness of one's present), they occur in a wide variety of contexts. In these, time itself is not the predominant aspect of one's experience.

*Time-specific agentive feelings.* In contrast, we identified several time-specific phenomena that could be interpreted as more extreme variations of the aforementioned general feelings with a particular focus on time itself. Some people have reported **feeling stuck in the present** (e.g., "*Being so spatially isolated inside, cut off from others except by Zoom, also made me feel temporally cut off from times before or after the pandemic.*") or, in contrast, **feeling out of touch with the present** (e.g., "*I feel like time is moving on without me*"; "*The idea of work (sitting down and looking at the screen) is eaten up by the patterns of the day (what I call the whole of time). The screen and the work seem without substance and not part of time.*").

These feelings can escalate with **time feeling standing still** (e.g., "*It's especially since the curfew, feeling that time stops at 6pm and no motivation to continue the tasks I have to continue.*") and **time feeling wasted** (e.g., "*The days go by at a pretty crazy pace, which adds to my feeling that this is so much 'wasted' time.*"). The most severe phenomenon we noticed was a form of depersonalization proper, with **time feeling unreal** (e.g., "*What day is it? What time is it? What should I do? I could not answer these questions. This period seems to me to be like a big void, a moment when nothing mattered anymore, I was in a black hole during these few months.*"). Depersonalization (or derealization) is a commonly reported organism response in traumatic settings [53] and has been documented during Covid-19 pandemic, particularly in traumatically exposed populations such as healthcare workers [54]. Similar to the previously discussed hopelessness, our reports indicate that some of the most extreme experiences during the Covid-19 restriction periods can be framed as traumatic events.

## Disorientation-related states

It is important to distinguish between disorientation and objectively being lost. For example, one might be lost when thinking it is 9AM when it is actually 10 AM. However, disorientation occurs when one realizes they are an hour late for a scheduled video conference call. Crucially, disorientation is an agentive feeling: it occurs when one cares about getting

oriented in the first place. Accordingly, some people from our corpus were **temporally lost** but not disoriented (e.g., "*I have personally experienced a certain suspension of calendar time -- I never seem to know what day it is, for it doesn't matter since I'm essentially housebound*") while others were first temporally lost, then experienced **episodic disorientation** (e.g., "*Sometimes I neglect to check my calendar, in times of confinement (because I usually don't have anything important to do during the day), so I forget an appointment that is there*"; "*It felt like time stood still for many months while life was 'on hold' and every day was similar, with no punctuation by landmark events. So I felt unready for Christmas to come around when it did.*").

A second key distinction that we identified in our reports are instances of experienced disorientation which are not episodic, as they more closely can be defined as **sustained temporal disbelief** (e.g., "*Like now for example I still can't realise that it's March already.*"; "*I'm shocked that it's been a year already*";"*I did not feel a loss of temporal reference points, but I had the feeling of a stretching of time that seemed abnormal and against which I had to fight without knowing how.*"). Unlike episodic disorientation, sustained temporal disbelief occurs when one knows the current time, yet the temporal perspective feels strange and uncanny. This form of disorientation persists even after one becomes temporally oriented, hence the term 'sustained'. Crucially, sustained and episodic forms of disorientation differ in their phenomenological structure. Episodic disorientation resembles metacognitive feelings, such as the tip-of-the-tongue phenomenon (where we feel we're about to recall a known word) [55]. In contrast, sustained disorientation is more akin to an existential feeling, representing an altered sense of belonging to the world and reality [56].

### Interpreting multiple correspondence analysis

The variables cluster in three different directions, which we interpret as episodic disorientation, sustained temporal disbelief, and future-oriented anxiety ([Fig 2]).

In the top-left area, "being lost in time", "difficulties to identify the current time", "week-scale" and "episodic disorientation" stand out. These variables are in the same direction as the more representationally fuzzy "days week weeks months looking the same" as well as "time feeling eventless". This alignment is consistent with our earlier observation that episodic disorientation often follows being temporally lost due to impaired time tracking, supporting our conceptualization of the "alterations of temporal orientation" theme. At the pandemic's onset, temporal disorientation manifested straightforwardly in the episodic form. Spatial and social confinement measures made it significantly more difficult to monitor and rely on spatially and socially distributed temporal landmarks that typically aid our orientation. This description is corroborated by the unit of meaning most extremely located in this cluster by MCA: "*During the confinement: the feeling of starting the same day again, every day. This cyclical aspect of my week - which wasn't really a week anymore, because I often forgot what day it was - I ended up adopting it myself in my personal organization: I had breakfast, had a coffee, did sports, and worked, almost always in that order. I know we often talk about daily routine, but it is usually imposed on us by schedules, it is the consequence of certain social imperatives. During the confinement, I voluntarily set it for myself, and I think that it was surely a way to have a control on the events, without letting myself be completely caught up in this extended time.*".

On the right of MCA, we observe several clustered variables, including "longer temporal distances", "past", "before covid", "shorter temporal distances", "events or periods feeling incoherently ordered", "difficulties to remember events or a period", "time feeling unreal", "sustained temporal disbelief" and "year-scale". This clustering aligns with our earlier observation that sustained temporal disbelief typically stems from an uncanny and confusing past-oriented temporal perspective. The most distinctive theme was "subjective temporal distances", and MCA reveals it was mostly associated with a year-scale frame of reference. Experientially, this distorted past perspective could manifest as a form of depersonalization, where time feels unreal and one remains in a state of existential disbelief. This is evidenced by the following reports located in this cluster: "*I often think of 2019 as "last year", 2020 seems not to have existed, like a blank, passed by very quickly, without having left any marks.*"; "*The more it goes on, the more I have the impression that the pre-covid period*

*belongs to an ancestral past remembered by oral tradition and a latent nostalgia rather than to a lived past and a living personal memory.*"; "*An event that happened last month feels both like yesterday and ten years ago.*"; "*The period of the summer seems to me much more distant, ancient than the first confinement.*".

In the the bottom left quadrant, "hopelessness" emerges as the most extreme variable. It is closely associated with "feeling stuck in the present", "difficulties to project oneself in the future", "future", "anxiety". To a lesser extent, it is also associated with "difficulties to imagine a different state of affairs". Two themes are the driving component of this cluster: "general agentive feelings" and "alterations of mental time travel". These themes are clearly exemplified by units of meaning in this cluster: "*Feeling of being a prisoner. Loss of normal rhythm, paralysis. This prevents one from acting and organizing oneself.*"; "*The future seems absolutely unattainable, and the impression of being stuck in an unpleasant perpetual present where nothing will ever have to happen is tenacious.*"; "*It has become very difficult to have a future. If it really depressed me for a while (between November 2020 and January 2021, I felt like our future as young people was totally stolen from us in terms of the consequences we're going to face for a long time, in addition to not being able to enjoy what should be the coolest years of our lives), I finally managed to deal with it and put it into perspective. I don't plan anything, to avoid being disappointed, and I hope it will pass one day.*"

MCA reveals that the two previously described forms of disorientation (present episodic confusion and past-oriented sustained disbelief) are insufficient to fully describe the phenomenon. We identify a missing third form. Three reasons support this claim. Firstly, hopelessness and anxiety are leading phenomenological features of *spatial* disorientation [57]. These features are closely associated in this cluster, and no other disorientation-related tags are present. This suggests our previous characterization of temporal disorientation is incomplete. Secondly, as a counterfactual addendum of the first point, the meta-theme "disorientation related states" is slightly off-center in the upper part of the MCA graph. If we had exhaustively covered all forms of disorientation encompassing the complete experiential disruptions, this meta-theme should have been centrally located. Its off-center position suggests an additional form of disorientation in the lower part of the graph. Thirdly, this type of disorientation is future-oriented, as evidenced by the high frequency (21%) of respondents reporting difficulties in projecting themselves into the future. One is trying to feel the presence of future courses of events but can't help but feel that these are not there on one's future timeline. Interestingly, while time "feeling unreal" was identified as the agentive hallmark of the past-oriented disbelief cluster, "hopelessness" appears to be its equivalent in the future-oriented disorientation cluster.

Our MCA reveals three distinct forms of temporal disorientation that structured temporal experience during the Covid-19 restriction era. These three forms of temporal disorientation not only correspond to the three temporal directions, but they also point us to three overarching types of temporal disruption: present-oriented temporal disorientation, which was mostly related to alterations in temporal orientation and was phenomenologically episodic; past-oriented temporal disorientation, which was primarily grounded in feelings of subjective temporal distances and induced sustained disbelief; future-oriented temporal disorientation, which was characterized by projection impairments and agentive feelings such as anxiety and hopelessness.

## Limitations and future directions

While the aim of our study is to document as accurately as possible the subjective disruptions that people reported during restrictions in the Covid-19 pandemic era, its generalizability is limited. Primarily, our sample was limited and recruited via convenience sampling, focusing on higher education students in France and the UK. The subjective experience of the same restrictions among different sociodemographic groups may reveal different disruption patterns. Secondly, the disorientation structure we identified, mapping typical disorienting disruptions to each of the temporal directions, may not generalize well to other disorienting situations, such as wars or catastrophes. Future studies could aim to consolidate reports from various crises and identify patterns that appear robust across different events. Finally, temporal disorientation, while a potent theoretical framework to approach the phenomenology of crisis, would benefit from further empirical

studies investigating its psychometric factor structure. This would allow the concept to be better operationalized in quantitative studies.

## Conclusion

Due to the various unexpected, long-lasting, and wide-ranging restrictions during Covid-19, the people from our sample felt temporally disoriented in three distinctive ways: present episodic confusion, past-oriented sustained disbelief, and future-oriented anxiety.

The sudden adoptions of restriction measures during the various epidemic outbreaks caused people to lose their usual temporal landmarks. This led to difficulties in monitoring time throughout their weekly schedule, which resulted in people getting lost in time. Consequently, some of them experienced episodic (though momentary) disorientation. Conversely, as restrictions evolved over more than a year, some of our participants' perspective on the past appeared confusing to them. This led to a form of disorientation best characterized by sustained disbelief and depersonalization. From a phenomenological perspective, some researchers have argued that catastrophes disrupt our natural agentive inclination for future-orientation, causing our lived temporal experience to "bring us face to face with contingency" [58]. Of concern, the widely documented international rise in anxiety and hopelessness during this period coincided with people's inability to project themselves into the future, resulting in a sense of disorientation.

Our analysis illuminates three overarching forms of temporal disorientation. However, both our comprehensive taxonomy and the relatively low explained inertia captured by multiple correspondence analysis emphasize the important variability in individual experiences of the crisis. These experiential disruptions of lived time are underpinned by a wide variety of cognitive functions and associated feelings. More work is needed to understand how these affective aspects of temporal experiences can be integrated into holistic frameworks, such as temporal disorientation [59].

We concur with other researchers (e.g., [60]) that changes in temporal perspectives are one of the most interesting variables to monitor in the general public in instances of severe societal crises (e.g., temporal perspectives during Covid-19 are significant predictors of depression and anxiety in [61]; or compliance with restrictions in [62] and in [63]). Regarding crisis management, the present analysis suggests taking actions to provide safer and more certain future landmarks during crises. These landmarks could serve as anchors for those most vulnerable to temporal disorientation.

## Author contributions

**Conceptualization:** Bastien Perroy, Pablo Fernandez Velasco, Roberto Casati.

**Data curation:** Bastien Perroy, Umer Gurchani.

**Formal analysis:** Bastien Perroy, Pablo Fernandez Velasco, Umer Gurchani.

**Funding acquisition:** Pablo Fernandez Velasco.

**Investigation:** Bastien Perroy, Pablo Fernandez Velasco.

**Methodology:** Bastien Perroy, Pablo Fernandez Velasco.

**Project administration:** Bastien Perroy, Pablo Fernandez Velasco.

**Resources:** Bastien Perroy.

**Supervision:** Pablo Fernandez Velasco, Roberto Casati.

**Validation:** Pablo Fernandez Velasco.

**Visualization:** Bastien Perroy.

**Writing – original draft:** Bastien Perroy, Pablo Fernandez Velasco.

**Writing – review & editing:** Bastien Perroy, Pablo Fernandez Velasco, Umer Gurchani, Roberto Casati.

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
