## [Decision Letter · Decision Letter 0]

Dear Dr. Fernandez Velasco,

Thank you for submitting your manuscript to PLOS ONE. After careful consideration, we feel that it has merit but does not fully meet PLOS ONE’s publication criteria as it currently stands. Therefore, we invite you to submit a revised version of the manuscript that addresses the points raised during the review process.

The manuscript has been evaluated by two reviewers, and their comments are available below.

The reviewers have raised a number of  concerns. They feel the methods section of the manuscript should be improved with further details, particularly regarding the study design and sample size determination. Additionally, I would recommend to discuss your findings in relation with more recent literature.

Could you please carefully revise the manuscript to address all comments raised?

*Comments from PLOS Editorial Office: We note that one or more reviewers has recommended that you cite specific previously published works. As always, we recommend that you please review and evaluate the requested works to determine whether they are relevant and should be cited. It is not a requirement to cite these works. We appreciate your attention to this request.*

We look forward to receiving your revised manuscript.

Kind regards,

Johanna Pruller, Ph.D.

Associate Editor

PLOS ONE

Journal Requirements:

2. Please include a separate caption for each figure in your manuscript.

Reviewers' comments:

Reviewer's Responses to Questions

**Comments to the Author**

1. Is the manuscript technically sound, and do the data support the conclusions?

Reviewer #1: Yes

Reviewer #2: Yes

2. Has the statistical analysis been performed appropriately and rigorously?

Reviewer #1: Yes

Reviewer #2: Yes

3. Have the authors made all data underlying the findings in their manuscript fully available?

Reviewer #1: Yes

Reviewer #2: Yes

4. Is the manuscript presented in an intelligible fashion and written in standard English?

Reviewer #1: Yes

Reviewer #2: Yes

Reviewer #1: Dear authors,

I enjoyed reading your manuscript and I think that your research is very interesting and focuses on a scarcely studied topic.

Objectives: The manuscript is interesting and pursues an original objective: to investigate the time experience in time of crisis (COVID-19 crisis) in general population by narratives of general population.

Method. The method includes the thematic analysis of 149 subjective reports gathered in March 2021 and the use of multiple correspondence analysis to resume the qualitative data. It is not clear if the authors use any qualitative software for the analysis, but I positively value the use of Sankey diagrams to visualize the results, as well as the guarantees for the analysis (intercoder reliability -Cohen’s κ as well as Krippendorff’s α-, blind revision, triangulation of methods for data analysis, and consensus among specialists for content analysis). I also value the use of the McNemar's test of marginal homogeneity and phi correlation coefficient for assessing significant relationships and performing Multiple Correspondence Analysis (MCA) for assess the dimensionality in categorical variables.

Results: The main results include three major topics: present episodic confusion, past-oriented sustained disbelief, and future-oriented anxiety. Although the reported temporal disorientation was evident during the lockdown, it has also been reported in times of economic and political crises: for example, the studies by Sautú & Flaherty on the distortion of time in Argentina (https://doi.org/10.1177/019027250506800407). Moreover, the notion of time is quickly lost even in non-crisis moments, such as during vacations. I also find the results related to the present intriguing, as they are actually the least clear (Table 3). As William James said, the past is memory and the future is speculation, but what is the present? In studies we are currently conducting on what people consider to be the past, present and future, the narratives are rich regarding the past and future but very poor and erratic regarding the present.

Conclusion: The conclusions are clear and consistent with the proposed objectives and methods. However, I believe that the limitations of the study should be indicated, as well as the future lines of research.

In summary, this is an interesting manuscript with an original purpose, good methodology and relevant results.

The quality of the writing and language is correct, and the manuscript is understandable. Although it will surely improve with the revision of a native English speaker.

Regarding the title, I suggest to avoid mentioning methods in it.

Reviewer #2: - The topic discusses temporal disorientation and the COVID-19 crisis, and I have some comments to improve the manuscript;

-In the introduction, the authors should begin by providing an update on the COVID-19 pandemic, including the spread of cases within and between waves worldwide, and then examine populations with the latest information. The following references could help enhance the first part of the introduction:

- https://doi.org/10.1371/journal.pone.0277368

- https://doi.org/10.3390/healthcare11172418

- https://doi.org/10.3390/healthcare10101858

- The introduction first highlights the general impacts of COVID-19 on different sectors before specifically addressing the psychological effects. An updated study further emphasizes this point, highlighting the same aspects in addition to the above references.

- https://doi.org/10.3390/children10111742

- https://doi.org/10.1016/j.sleepe.2022.100030

- https://doi.org/10.1007/s11469-024-01297-x

- https://doi.org/10.1136/bmjopen-2020-046006

- https://doi.org/10.3390/ejihpe12080079

Under the methodology section, please include and explain the following subsections:

1. Study design - Provide a detailed explanation of the chosen study design and the reasons for using it.

2. Sample size and sampling procedures - Explain how the sample size was determined and describe the sampling methods used.

3. Instruments - Provide detailed explanations regarding the reliability and validity of the instruments used in the study, along with information on their scores. If an adopted scale or a created scale is used, provide details on how it was validated.

- Please ensure that the methodology section includes a detailed explanation of the statistical analysis. Additionally, in the statistical section, please provide a thorough explanation of the tests performed.

- Please discuss the theoretical and practical implications of your study findings in the Discussion section.

**Do you want your identity to be public for this peer review?** For information about this choice, including consent withdrawal, please see our Privacy Policy

Reviewer #1: **Yes: ** María Elena Brenlla Victoria

Reviewer #2: No

---

## [Author Response · Author response to Decision Letter 1]

16 Aug 2024

Dear editorial team, dear reviewers,

Thank you so much for this encouraging and thorough review. We have addressed all the recommendations you provided. Most notably, we have restructured the methods section to provide more details about the study design, limitations, and future directions. We have also conducted a thorough additional proofreading of our manuscript for idiomaticity, leading to numerous minor revisions. We've added a statistical correction for multiple comparisons to enhance the reliability of our results. We have revised the introduction and discussion sections to include recent and influential studies on this topic. These include Alatrany et al., 2022, Kosak et al., 2022, and Ogden and Piovesan, 2022, among others. We believe these changes have substantially improved our manuscript.

We include the original comments from reviewers below in grey font, and our reply in black font (see attached response to reviewers document).

Reviewer #1: Dear authors,

I enjoyed reading your manuscript and I think that your research is very interesting and focuses on a scarcely studied topic.

Objectives: The manuscript is interesting and pursues an original objective: to investigate the time experience in time of crisis (COVID-19 crisis) in general population by narratives of general population.

Method. The method includes the thematic analysis of 149 subjective reports gathered in March 2021 and the use of multiple correspondence analysis to resume the qualitative data. It is not clear if the authors use any qualitative software for the analysis, but I positively value the use of Sankey diagrams to visualize the results, as well as the guarantees for the analysis (intercoder reliability -Cohen’s κ as well as Krippendorff’s α-, blind revision, triangulation of methods for data analysis, and consensus among specialists for content analysis).

We are grateful for this overall assessment of our manuscript. We have now mentioned our use of Excel as qualitative software. This choice was due to the need for complex multi-labelling and negotiation logic, which was not readily available in other software packages. All statistical tests and our data are detailed in sheets available in the OSF data archive.

I also value the use of the McNemar's test of marginal homogeneity and phi correlation coefficient for assessing significant relationships and performing Multiple Correspondence Analysis (MCA) for assess the dimensionality in categorical variables.

Results: The main results include three major topics: present episodic confusion, past-oriented sustained disbelief, and future-oriented anxiety. Although the reported temporal disorientation was evident during the lockdown, it has also been reported in times of economic and political crises: for example, the studies by Sautú & Flaherty on the distortion of time in Argentina (https://doi.org/10.1177/019027250506800407). Moreover, the notion of time is quickly lost even in non-crisis moments, such as during vacations. I also find the results related to the present intriguing, as they are actually the least clear (Table 3). As William James said, the past is memory and the future is speculation, but what is the present? In studies we are currently conducting on what people consider to be the past, present and future, the narratives are rich regarding the past and future but very poor and erratic regarding the present.

Thank you for your comment and for directing us to the relevant Sautú & Flaherty study, which we have now referenced. Indeed, temporal disorientation can be observed not only during lockdowns but also in various other socio-political crises. We have now expanded our introduction to better address these additional contexts. We concur with your observation that there is a scarcity of studies comparing distortions in future and past perspectives with those in the present. We hope our work contributes to this discussion, and we're pleased that you found this line of thought stimulating.

Conclusion: The conclusions are clear and consistent with the proposed objectives and methods. However, I believe that the limitations of the study should be indicated, as well as the future lines of research.

We thank you for this feedback. We now conclude our discussion section with a "Limitations and Future Directions" subsection. We highlight three main limitations of our study:

- Limited generalizability across sociodemographic groups due to our sampling procedure.

- The need to compare our findings with reports from other types of crises, such as wars and natural catastrophes.

- The necessity for more thorough psychometric investigation into the various ways the concept of disorientation can be operationalized when researching the psychological impact of crises.

In summary, this is an interesting manuscript with an original purpose, good methodology and relevant results. The quality of the writing and language is correct, and the manuscript is understandable. Although it will surely improve with the revision of a native English speaker.

We have conducted an additional, thorough proofreading of the manuscript to improve its idiomaticity. This resulted in numerous minor revisions throughout the text.

Regarding the title, I suggest to avoid mentioning methods in it.

Thank you for your feedback. We have revised our title to: “Three Forms of Temporal Disorientation: A Thematic Analysis of Subjective Reports During Covid-19 Restriction Periods”.

We would like to express our sincere gratitude to Reviewer #1 for the detailed and constructive feedback, which reflects a thorough engagement with our manuscript. We believe these insights have significantly enhanced the quality of our work.

#######

Reviewer #2: - The topic discusses temporal disorientation and the COVID-19 crisis, and I have some comments to improve the manuscript;

-In the introduction, the authors should begin by providing an update on the COVID-19 pandemic, including the spread of cases within and between waves worldwide, and then examine populations with the latest information. The following references could help enhance the first part of the introduction:

- https://doi.org/10.1371/journal.pone.0277368

- https://doi.org/10.3390/healthcare11172418

- https://doi.org/10.3390/healthcare10101858

- The introduction first highlights the general impacts of COVID-19 on different sectors before specifically addressing the psychological effects. An updated study further emphasizes this point, highlighting the same aspects in addition to the above references.

- https://doi.org/10.3390/children10111742

- https://doi.org/10.1016/j.sleepe.2022.100030

- https://doi.org/10.1007/s11469-024-01297-x

- https://doi.org/10.1136/bmjopen-2020-046006

- https://doi.org/10.3390/ejihpe12080079

Thank you for providing these references. We were particularly interested in the findings of Aljaberi et al. (2023), which reported that approximately 36% of their sample exhibited PTSD symptoms. We have now included this reference in our manuscript. Additionally, we have referenced the earlier study by Aljaberi et al. (2022) on the interactions between anxiety, depression, and insomnia during the pandemic. These factors have also been discussed in the time perception literature, and we now highlight that these studies converge on the view that further research is necessary to address the multifaceted aspects of psychological health during crises. While we are mindful of not over-expanding the paper with numerous new references, we believe that the introduction is now more focused, thanks to your insightful suggestion.

Under the methodology section, please include and explain the following subsections:

1. Study design - Provide a detailed explanation of the chosen study design and the reasons for using it.

Thank you for your comment. We have retitled the first subsection of the methods section to “Study design.” This section now offers a more seamless introduction, including additional references, which clarify the rationale for combining thematic analysis with multiple correspondence analysis.

2. Sample size and sampling procedures - Explain how the sample size was determined and describe the sampling methods used.

We appreciate this comment, which identified an area of our methods section that required further detail. We have now retitled the second subsection of the methods section to “Sampling Procedure” and have thoroughly revised its content. In this revised subsection, we explicitly state that we employed a convenience sampling strategy, along with the rationale behind this choice. We have also expanded on the rationale for our sample size, aiming to balance the manageability of the coding process with the goal of detecting strong associations. As a result, this subsection has undergone significant revisions. Additionally, we have included a new “Limitations and future directions” subsection at the end of the discussion, where we address the sampling procedure as a key factor that limits the generalizability of our results.

3. Instruments - Provide detailed explanations regarding the reliability and validity of the instruments used in the study, along with information on their scores. If an adopted scale or a created scale is used, provide details on how it was validated.

Thank you for raising this question regarding reliability and validity. As our study employed a qualitative approach utilizing an open-ended questionnaire, it does not rely on the standardized scales typically associated with quantitative instruments. However, we adhered to widely documented best practices for qualitative research:

- To minimize bias in question wording, we drew upon existing questionnaires from health studies (see the newly titled “Study design” section).

- To ensure reliability, each unit of meaning was independently coded by two coders in multiple rounds (details are provided in the newly titled “Reliability assessment: a 3-step tagging process” subsection). We employed Cohen's κ and Krippendorff's α, standard metrics in the literature for intercoder reliability, due to their ability to account for chance agreement.

- We applied McNemar's test for significance testing, providing a detailed rationale for its selection.

To address these inquiries more effectively, the “Study design” subsection now explicitly introduces our reliance on thematic analysis and references both a methodologically similar study published in PLOS One on dementia, and one of Braun and Clarke's influential papers on thematic analysis methods. We also renamed subsections to clarify their objectives (e.g., “Study design”, “Sampling procedure”, “Coding procedure”, “Reliability assessment”, “Statistical tests”, “Interpretative instrument”, etc.).

While our approach has inherent limitations, including the potential for subjectivity in qualitative coding, we have mitigated these through a rigorous multi-coder process and statistical validation of inter-coder reliability. We recognize that our findings may not be as generalizable as those from large-scale quantitative studies. This issue is now addressed in detail in a new subsection titled “Limitations and future directions”.

- Please ensure that the methodology section includes a detailed explanation of the statistical analysis. Additionally, in the statistical section, please provide a thorough explanation of the tests performed.

We appreciate your challenge regarding our statistical analysis, which prompted us to strengthen the corresponding subsections.

We believe the titled subsection “Statistical tests: using McNemar's Test to assess non-marginal homogeneity” in the Methods section already provided a thorough explanation for our choice of statistical analysis but lacked clarity in certain areas.

Nemar's test was selected to assess whether the marginal frequencies of our paired nominal data were equal (i.e., to test for non-marginal homogeneity), a crucial step in evaluating associations between paired categorical variables. An exact test was required due to the high number of contingency tables (1,114 in total, representing all possible pairwise comparisons of our coded themes and tags) and the presence of low expected frequencies in some tables, rendering the Chi-square test of independence unsuitable. Additionally, our paired data made Fisher's exact test inappropriate. We have also elaborated on the existence of several versions of McNemar's test and cited a meta-analysis methodology paper that supports our use of the mid-p version. The revised wording in this subsection now clarifies these points.

Initially, we analyzed only the associations found to be significant (with an alpha threshold of .05) in the Results section, reporting the p-value for each. To address the issue of multiple comparisons, we have now applied the Benjamini-Hochberg correction to the tests included in our data archive and provided a rationale for this choice in the Methods section. Consequently, all correlations detailed in the Results section remain significant after applying the Benjamini-Hochberg correction, ensuring a low risk of false positives.

In response to your query, we have now indicated that all statistical tests performed are available in the data archive associated with our study. We have also included a summary of the methods at the beginning of the Results section. Additionally, we have integrated some of the metrics from the figure into the text to enhance readability. The raw data in our data archive enables the reproducibility of the statistical tests detailed in the Results section.

- Please discuss the theoretical and practical implications of your study findings in the Discussion section.

We appreciate your comment, which has prompted us to reflect on the implications of our study's findings. In response, we have expanded our discussion to more explicitly address both the practical and theoretical implications.

Concerning practical implications, our conclusion already outlines several policy recommendations: monitoring shifts in temporal perspectives at the population level using relevant indicators, and ensuring that individuals most vulnerable to temporal disorientation have access to future temporal landmarks during crises. Due to the qualitative nature of our analysis, we remain cautious about extending our practical recommendations further.

The theoretical implications of our study were not sufficiently detailed in our original submission. To rectify this, we have added a “Limitations and Future Directions” subsection to the discussion. In this new section, we address two key theoretical implications that could be valuable for future research:

- Generalizability across crisis types: For our thematic analysis results to be generalizable, similar patterns must be identified in future studies investigating other disorienting situations, such as wars or natural disasters. Such cross-crisis comparisons could validate or refine the proposed structure of temporal disorientation.

- Operationalization of ‘disorientation’: While ‘disorientation’ is a potent concept for analyzing crises, its psychometric factor structure requires further investigation. Additional research could enable its operationalization in large-scale quantitative instruments, potentially advancing our understanding of psychological responses to crises.

These additions to our discussion underscore the potential impact of our findings and identify key areas where further research is warranted.

We extend our gratitude to Reviewer #2 for his feedback on our manuscript.

---

## [Decision Letter · Decision Letter 1]

Dear Dr. Fernandez Velasco,

We look forward to receiving your revised manuscript.

Kind regards,

Vincenzo De Luca

Academic Editor

PLOS ONE

Reviewers' comments:

Reviewer's Responses to Questions

**Comments to the Author**

Reviewer #1: All comments have been addressed

Reviewer #2: (No Response)

2. Is the manuscript technically sound, and do the data support the conclusions?

Reviewer #1: (No Response)

Reviewer #2: Partly

3. Has the statistical analysis been performed appropriately and rigorously?

Reviewer #1: (No Response)

Reviewer #2: I Don't Know

4. Have the authors made all data underlying the findings in their manuscript fully available?

Reviewer #1: (No Response)

Reviewer #2: No

5. Is the manuscript presented in an intelligible fashion and written in standard English?

Reviewer #1: (No Response)

Reviewer #2: No

Reviewer #1: (No Response)

Reviewer #2: Please make sure to include the following information in the introduction:

- Start the Introduction by discussing the current status of the COVID-19 pandemic and its global impact, citing specific examples of case spread within and between waves, as in the following references and you can use also more update studies in addition to this;

https://doi.org/10.3390/healthcare10101858

- Provide an overview of the pandemic's effects on various sectors before delving into its psychological impact. Support your points with the following references:

Additionally, refer to the following updated study for further insight:

https://doi.org/10.3390/children10111742

https://doi.org/10.1016/j.sleepe.2022.100030

https://doi.org/10.3390/ejihpe12080079

- at the end of the discussion add the following sections and explained it;

Theoretical and practical implications of the study, and Strength and limitations of the study

**Do you want your identity to be public for this peer review?** For information about this choice, including consent withdrawal, please see our Privacy Policy

Reviewer #1: **Yes: ** María Elena Brenlla

Reviewer #2: No

---

## [Author Response · Author response to Decision Letter 2]

10 Dec 2024

Dear Editor,

We are writing regarding our manuscript currently under review, as we understand you have recently taken over its editorial handling.

The primary outstanding issue from the latest review round concerns Reviewer 2's request for additional citations. In the initial review, R2 suggested including eight papers authored by Musheer A. Aljaberi. After thorough evaluation, we incorporated the two most relevant citations. In the subsequent review, R1 accepted the manuscript, but R2 requested the inclusion of four additional papers from the same author. Upon careful consideration, we find these additional citations do not substantially contribute to our manuscript's scholarly content.

For context, R2's second-round comments appear to overlook our major revisions, instead reiterating points from their initial review. For instance, their request for a limitations section disregards that this was already implemented in our revision. Given that Reviewer 1 has expressed satisfaction with our major revision, we have decided to maintain our manuscript in its current form. Our point-by-point responses to R2's outstanding requests are provided below.

Sincerely,

Pablo Fernandez Velasco and Bastien Perroy

6. Review Comments to the Author

Reviewer #1: (No Response)

Reviewer #2: Please make sure to include the following information in the introduction:

- Start the Introduction by discussing the current status of the COVID-19 pandemic and its global impact, citing specific examples of case spread within and between waves, as in the following references and you can use also more update studies in addition to this;

https://doi.org/10.3390/healthcare10101858

- Provide an overview of the pandemic's effects on various sectors before delving into its psychological impact. Support your points with the following references:

Additionally, refer to the following updated study for further insight:

https://doi.org/10.3390/children10111742

https://doi.org/10.1016/j.sleepe.2022.100030

https://doi.org/10.3390/ejihpe12080079

These comments were previously raised in your initial review. During our first revision, we incorporated additional citations to contextualize the global impact of the Covid-19 pandemic. We maintain our position from the previous round of reviews: additional citations regarding the global context would not add substantive value to our manuscript, which specifically focuses on the psychological and experiential toll of Covid-19 in the general population. For reference, our previous reply stated:

“We were particularly interested by Aljaberi et al. (2023) findings that around 36% of their sample had PTSD symptoms, which we now reference. We also reference the other study by Aljaberi et al. (2022) on the interactions between anxiety, depression, and insomnia during the pandemic. In the time perception literature, these factors have also been explored and we now mention that these results all converge in the view that more work is needed on the multifaceted aspects of psychological health during crises. We are cautious on not expanding the paper too much with too many new references but we believe the introduction is much more focused now, thanks to your comment.”

- at the end of the discussion add the following sections and explained it;

Theoretical and practical implications of the study, and Strength and limitations of the study

The theoretical and practical implications of the study, along with its strengths and limitations, were incorporated during the previous round of reviews. We maintain that these sections are sufficiently comprehensive and do not require further expansion. For your reference, we have included below our original explanation for these additions from the previous round:

“We have expanded our discussion to address both practical and theoretical implications more explicitly.

Regarding practical implications, our conclusion already outlines several areas for policy recommendations: monitoring shifts in temporal perspectives at the population level using indicators, and ensuring that individuals most vulnerable to temporal disorientation have access to as many future temporal landmarks as possible during times of crisis, most notably. Given the qualitative nature of our analysis, we are cautious about extending our practical recommendations much further.

The theoretical implications of our study were not adequately detailed in our original submission. To address this, we have added a “Limitations and future directions” subsection to the discussion. In this new subsection, we touch upon two key theoretical implications that may be valuable for future studies:

Generalizability across crisis types: For the results of our thematic analysis to be generalizable, similar patterns would need to be identified in future studies investigating other disorienting situations such as wars or natural disasters. This cross-crisis comparison could validate or refine our proposed structure of temporal disorientation.

Operationalization of ‘disorientation’: While “disorientation” proves to be a potent concept for analyzing crises, its psychometric factor structure requires further investigation. This additional research would enable its operationalization in large-scale quantitative instruments, potentially advancing our understanding of psychological responses to crises.

These additions to our discussion highlight both the potential impact of our findings and the areas where further research is needed.”

With respect to limitations, here was the explanation in the previous round of reviews:

“We now conclude our discussion section with a “Limitations and Future Directions” subsection. We mention 3 outstanding limitations: the limited generalizability across socio demographic groups due to our sampling procedure, the need to compare these reports to those that could be collected in other types of crises such as wars and catastrophes, and finally more thorough psychometric investigation on the various ways a concept such as disorientation may be operationalised for researching the psychological impact of crises.”

############################################################################

For reference, we are including our first round of reviews as well.

First round of reviews

Reviewer #1: Dear authors,

I enjoyed reading your manuscript and I think that your research is very interesting and focuses on a scarcely studied topic.

Objectives: The manuscript is interesting and pursues an original objective: to investigate the time experience in time of crisis (COVID-19 crisis) in general population by narratives of general population.

Method. The method includes the thematic analysis of 149 subjective reports gathered in March 2021 and the use of multiple correspondence analysis to resume the qualitative data. It is not clear if the authors use any qualitative software for the analysis, but I positively value the use of Sankey diagrams to visualize the results, as well as the guarantees for the analysis (intercoder reliability -Cohen’s κ as well as Krippendorff’s α-, blind revision, triangulation of methods for data analysis, and consensus among specialists for content analysis).

We are grateful for this overall assessment of our manuscript. We have now mentioned the use of Excel as our qualitative software due to the complex multi-labelling and negotiation logic that needed to be implemented (and which was not readily available in the other softwares we had access to). Sheets are still available in the OSF archive detailing all statistical tests as well as disclosing our data.

I also value the use of the McNemar's test of marginal homogeneity and phi correlation coefficient for assessing significant relationships and performing Multiple Correspondence Analysis (MCA) for assess the dimensionality in categorical variables.

Results: The main results include three major topics: present episodic confusion, past-oriented sustained disbelief, and future-oriented anxiety. Although the reported temporal disorientation was evident during the lockdown, it has also been reported in times of economic and political crises: for example, the studies by Sautú & Flaherty on the distortion of time in Argentina (https://doi.org/10.1177/019027250506800407). Moreover, the notion of time is quickly lost even in non-crisis moments, such as during vacations. I also find the results related to the present intriguing, as they are actually the least clear (Table 3). As William James said, the past is memory and the future is speculation, but what is the present? In studies we are currently conducting on what people consider to be the past, present and future, the narratives are rich regarding the past and future but very poor and erratic regarding the present.

Thank you for this comment and hinting us towards the relevant Sautú & Flaherty study, which we now reference. It is true that temporal disorientation can be observed not only during lockdowns but a number of other socio-political crises, and our manuscript better hints at these other situations in the introduction. We agree with you that too few studies compare distortions to future and past perspectives with those that can be contrasted with in the present. We hope to contribute to the discussion and are happy to see that you found this line of thoughts stimulating.

Conclusion: The conclusions are clear and consistent with the proposed objectives and methods. However, I believe that the limitations of the study should be indicated, as well as the future lines of research.

We thank you for this feedback. We now conclude our discussion section with a “Limitations and Future Directions” subsection. We mention 3 outstanding limitations: the limited generalizability across socio demographic groups due to our sampling procedure, the need to compare these reports to those that could be collected in other types of crises such as wars and catastrophes, and finally more thorough psychometric investigation on the various ways a concept such as disorientation may be operationalised for researching the psychological impact of crises.

In summary, this is an interesting manuscript with an original purpose, good methodology and relevant results.

The quality of the writing and language is correct, and the manuscript is understandable. Although it will surely improve with the revision of a native English speaker.

We have now performed an additional and thorough proofreading of the manuscript for idiomaticity. This led to a very important number of minor revisions all throughout the text.

Regarding the title, I suggest to avoid mentioning methods in it.

Thank you for this feedback. We have now reformulated our title to the following: “Three Forms of Temporal Disorientation: A Thematic Analysis of Subjective Reports about Covid-19 Restriction Periods”

Again, we are very grateful to Reviewer #1 for detailed and constructive feedback, indicative of the time spent thoroughly engaging with our manuscript. Thanks to this feedback, we believe our manuscript has improved substantially.

Reviewer #2: - The topic discusses temporal disorientation and the COVID-19 crisis, and I have some comments to improve the manuscript;

-In the introduction, the authors should begin by providing an update on the COVID-19 pandemic, including the spread of cases within and between waves worldwide, and then examine populations with the latest information. The following references could help enhance the first part of the introduction:

- https://doi.org/10.1371/journal.pone.0277368

- https://doi.org/10.3390/healthcare11172418

- https://doi.org/10.3390/healthcare10101858

- The introduction first highlights the general impacts of COVID-19 on different sectors before specifically addressing the psychological effects. An updated study further emphasizes this point, highlighting the same aspects in addition to the above references.

- https://doi.org/10.3390/children10111742

- https://doi.org/10.1016/j.sleepe.2022.100030

- https://doi.org/10.1007/s11469-024-01297-x

- https://doi.org/10.1136/bmjopen-2020-046006

- https://doi.org/10.3390/ejihpe12080079

Thank you for sharing these references. We were particularly interested by Aljaberi et al. (2023) findings that around 36% of their sample had PTSD symptoms, which we now reference. We also reference the other study by Aljaberi et al. (2022) on the interactions between anxiety, depression, and insomnia during the pandemic. In the time perception literature, these factors have also been explored and we now mention that these results all converge in the view that more work is needed on the multifaceted aspects of psychological health during crises. We are cautious on not expanding the paper too much with too many new references but we believe the introduction is much more focused now, thanks to your comment.

Under the methodology section, please include and explain the following subsections:

1. Study design - Provide a detailed explanation of the chosen study design and the reasons for using it.

Thank you for your comment. The first subsection of the methods section is now titled ‘Study design’ and provides a smoother introduction, highlighting the rationale behind the choice of a thematic analysis (our study design), also referencing

2. Sample size and sampling procedures - Explain how the sample size was determined and describe the sampling methods used.

We are grateful for this comment, highlighting a part of our methods which was not as detailed as it could be. The second subsection in our methods section is now titled “Sampling procedure”. In this thoroughly revised subsection, we now explicitly state that we followed a convenience sampling strategy and provide the associated rationale. We also give more details on the sample size rationale, striking a balance between a manageable coding procedure and the aim of detecting strong associations throughout our coding process. Overall, this subsection has undergone major revisions. Additionally, in the new 'Limitations and future directions' subsection at the end of the discussion section, we also mention the sampling procedure as a key factor limiting the generalizability of our results.

3. Instruments - Provide detailed explanations regarding the reliability and validity of the instruments used in the study, along with information on their scores. If an adopted scale or a created scale is used, provide details on how it was validated.

Thanks for this question on reliability and validity. As our study employed a qualitative approach with an open-ended questionnaire, it does not rely on standardized scales typically found in quantitative instruments. Nonetheless, we followed best practices widely documented for such qualitative approaches:

- To minimize bias in question wording, we took inspiration from existing questionnaires in health studies (see the now-titled 'Study design' section).

- To ensure reliability, each unit of meaning was coded blindly by two coders through multiple coding rounds (more details in the section now titled “Reliability assessment: a 3-step tagging process”). We relied on Cohen's κ and Krippendorff's α, which are considered standard metrics in the literature for intercoder reliability due to their ability to account for chance agreement.

- We used McNemar's test for significance testing, detailing the precise rationale for its selection.

- Construct validity is more challenging to assess with indicators in qualitative analyses. We relied on independent note-taking for generating the taxonomic tree (see "Establishing a taxonomic tree and delimiting units of meaning") and made associations in the discussion section between reports illustrating each of these constructs and studies in the literature (see "Unfolding the taxonomic tree").

- Our thematic analysis approach, guided by Braun and Clarke's widely referenced methodology, contributes to the validity of our findings by ensuring a systematic and rigorous process of identifying, analyzing, and reporting patterns within the data.

To better account for such inquiries, the study design section now better introduces the fact that our

---

## [Decision Letter · Decision Letter 2]

Three Forms of Temporal Disorientation: A Thematic Analysis of Subjective Reports about Covid-19 Restriction Periods

PONE-D-23-40108R2

Dear Dr. Fernandez Velasco,

We’re pleased to inform you that your manuscript has been judged scientifically suitable for publication and will be formally accepted for publication once it meets all outstanding technical requirements.

Kind regards,

Avanti Dey, PhD

Staff Editor

PLOS ONE

Additional Editor Comments (optional):

Reviewers' comments:

Reviewer's Responses to Questions

**Comments to the Author**

Reviewer #1: All comments have been addressed

Reviewer #2: (No Response)

2. Is the manuscript technically sound, and do the data support the conclusions?

Reviewer #1: Yes

Reviewer #2: No

3. Has the statistical analysis been performed appropriately and rigorously?

Reviewer #1: Yes

Reviewer #2: No

4. Have the authors made all data underlying the findings in their manuscript fully available?

Reviewer #1: Yes

Reviewer #2: No

5. Is the manuscript presented in an intelligible fashion and written in standard English?

Reviewer #1: Yes

Reviewer #2: No

Reviewer #1: Dear authors,

I think that your paper provide interesting data about how people experienced the time under lockdown condition. These results will be useful for understand the passage of time in similar situations in the future.

With my bet regards,

Elena Brenlla

Reviewer #2: (No Response)

**Do you want your identity to be public for this peer review?** For information about this choice, including consent withdrawal, please see our Privacy Policy

Reviewer #1: **Yes: ** María Elena Brenlla

Reviewer #2: No

---

## [Editor Report · Acceptance letter]

PONE-D-23-40108R2

PLOS ONE

Dear Dr. Fernandez Velasco,

I'm pleased to inform you that your manuscript has been deemed suitable for publication in PLOS ONE. Congratulations! Your manuscript is now being handed over to our production team.

Kind regards,

on behalf of

Dr. Avanti Dey

Staff Editor

PLOS ONE